# Octopus: A Multi-modal LLM with Parallel Recognition and Sequential Understanding

**Chuyang Zhao**[1*]   **Yuxin Song**[1*]   **Junru Chen**[1]   **Kang Rong**[1]   **Haocheng Feng**[1]
**Gang Zhang**[1]   **Shufan Ji**[2]   **Jingdong Wang**[1]   **Errui Ding**[1]   **Yifan Sun**[1✉]

[1]Baidu VIS    [2] Beihang University
[*] Equal Contribution    [✉] Corresponding Author
{zhaochuyang,songyuxin02,sunyifan01}@baidu.com

## Abstract

A mainstream of Multi-modal Large Language Models (MLLMs) have two essential functions, *i.e.*, visual recognition (*e.g.*, grounding) and understanding (*e.g.*, visual question answering). Presently, all these MLLMs integrate visual recognition and understanding in a same sequential manner in the LLM head, *i.e.*, generating the response token-by-token for both recognition and understanding. We think unifying them in the same sequential manner is not optimal for two reasons: 1) parallel recognition is more efficient than sequential recognition and is actually prevailing in deep visual recognition, and 2) the recognition results can be integrated to help high-level cognition (while the current manner does not). Such motivated, this paper proposes a novel "parallel recognition $\rightarrow$ sequential understanding" framework for MLLMs. The bottom LLM layers are utilized for parallel recognition and the recognition results are relayed into the top LLM layers for sequential understanding. Specifically, parallel recognition in the bottom LLM layers is implemented via object queries, a popular mechanism in DEtection TRansformer, which we find to harmonize well with the LLM layers. Empirical studies show our MLLM named Octopus improves accuracy on popular MLLM tasks and is up to $5\times$ faster on visual grounding tasks.

## 1 Introduction

Visual recognition and understanding are two essential abilities for Multi-modal Large Language Models (MLLMs). While earlier MLLMs [1, 2, 3, 4] focused on the high-level visual understanding ability (*e.g.*, visual question answering), recent literature finds that visual recognition ability (*i.e.*, identifying and locating the objects) are no less important. The importance lies in two aspects: 1) Many newly-merged MLLM usages are directly related to visual recognition, *e.g.*, visual grounding [5, 6] and referential dialog [6]. 2) More generally, visual recognition is potential to benefit all understanding tasks as recognition results are important compositions for understanding. In this paper, we are interested in better harmonizing these two essentials for MLLM.

Presently, MLLMs unify visual recognition and understanding in a sequential paradigm. In this paper, the terms "sequential" and "parallel" refer specifically to the inference of LLM head, rather than the visual encoder. Typically, an MLLM consists of a visual encoder and an LLM head. During inference, the LLM head sequentially generates the response token-by-token, regardless of whether the task is more aligned with visual recognition (*e.g.*, grounding) or understanding (*e.g.*, visual question answering), as in Fig. 1 (left). Sequentially referring the recognition results, particularly textualized coordinates, is relatively slow. This sequential manner is a legacy of the LLM structure and, more fundamentally, stems from the inherently sequential nature of language.

38th Conference on Neural Information Processing Systems (NeurIPS 2024).

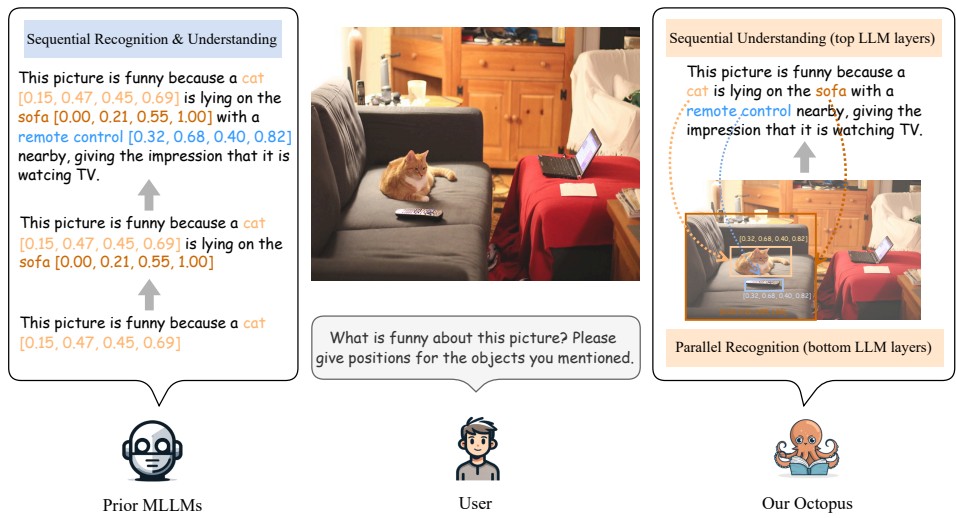

Figure 1: Comparison between prior MLLMs (left) and our Octopus (right). Left: Prior MLLMs typically adopt the purely sequential inference: the LLM head infers the response token-by-token, regardless whether the response token is more aligned with recognition (*e.g.*, detection) or understanding. Sequentially inferring the position is slow. Right: In contrast, Octopus establishes a "parallel recognition → sequential understanding" framework. The bottom LLM layers first recognize potential objects via visual grounding or referring segmentation (in Appendix:B). The recognition results (coupled with visual tokens) are relayed into top LLM layers. The top LLM layers thus do NOT infer the object position but instead, they select boxes (or masks) that have already been detected. The entire Octopus LLM head (recognition + understanding) is trained end-to-end.

We conjecture the purely sequential paradigm might not be an optimistic framework for MLLM, especially regarding visual recognition and its cooperation with understanding. There are two reasons:

● First, for both human and deep learning, visual recognition relies heavily on parallel processing for high efficiency. Before the MLLMs era, most deep recognition models are built on the parallel paradigm. For instance, the segmentation model infers the semantic for all pixels simultaneously, and the detection model detects all the objects using parallel anchors [7, 8] or queries [9, 10, 11, 12]. Humans also use parallel processing for simple recognition [13, 14, 15]. In contrast, current purely sequential paradigm lacks efficiency for visual recognition.

● Second, regarding the cooperation between visual recognition and understanding, there is a hierarchy of "parallel recognition → sequential understanding", as revealed by psychology and neurobiology discoveries [14, 15]. Relatively easier and low-level recognition results are integrated via more complex mental operations to help high-level cognition [13]. This hierarchy allows the understanding to take advantage of the recognition results, while the purely sequential paradigm does not offer such a benefit.

This paper proposes the Octopus (the octopus animal has a central brain and multiple parallel "auxiliary brains") framework to improve the efficiency of recognition, and to harness the benefit of the aforementioned cognition hierarchy. Octopus separates visual recognition and understanding into parallel and sequential processes, respectively, and then re-integrates them in a "parallel recognition → sequential understanding" hierarchy. The comparison between purely sequential MLLMs and Octopus is illustrated in Fig. 1.

Given visual tokens from the visual backbone, Octopus's LLM head uses the bottom layers to detect the potential objects in parallel. The detection results, coupled with the visual tokens, are fed into the sub-sequential LLM layers for further understanding. Though the understanding remains sequentially token-by-token, it deviates from previous MLLMs by eliminating the need to infer the position of objects. Instead, it selects previously-detected boxes and associates them with the objects, markedly improving efficiency. For example, on Flickr30k dataset (average 4 objects per image), Octopus reduces the time of recognizing all objects to about 21% (3.80s to 0.82s per image, 5× increase in speed). Moreover, we empirically find that Octopus improves the accuracy on a range of understanding tasks compared to its purely sequential counterpart. It indicates that the initial

parallel recognition can effectively support the understanding, revealing a clear advantage of the brained-inspired cognitive hierarchy.

Another significant feature of Octopus is: it can automatically adjust its recognition modes, oscillating between class-agnostic and class-specific, based on user instructions. This flexibility provides versatile usages for various tasks. For instance, in a grounding task where users specify the particular interest ,*e.g.*, a cat, the recognition part of Octopus becomes class-specific and and predicts multiple candidates for the object of interest. The understanding part then selects the best candidate for the final response. In contrast, in another scenario, where the users do not specify any particular interest and request a detailed enumeration (including the position), the recognition part automatically shifts into a class-agnostic detector. Correspondingly, the understanding part then assigns semantics to the detection results. This flexibility originates from the knowledge in the bottom LLM layers designated for recognition.

Our main contributions are as summarized as follows:

• We investigate the cooperation between recognition and understanding in current MLLMs. As a result, we identify an efficiency issue with the purely sequential paradigm, as well as a significant discrepancy from human cognitive processes.

• We propose the Octopus framework for MLLM. Octopus separates recognition and understanding into parallel and sequential processes, respectively, and re-integrates them into a "parallel recognition → sequential understanding" hierarchy.

• Extensive experiments show Octopus improves inference efficiency and enhances the accuracy, compared to the purely sequential counterpart.

## 2 Related Works

**Multi-modal Large Language Model.** The recent success of large language models (LLMs) has spurred research into integrating LLMs with computer vision for visual understanding. Flamingo [16] adds trainable cross-attention layers to each LLM decoder layer to learn visual information. BLIP-2 [3] introduces the Q-Former to align visual and language spaces. Mini-GPT4 [2] and mPLUG-OWL [17] also use the Q-Former for visual understanding. LLaVA [18] connects the pretrained CLIP [19] visual encoder to the LLM with a simple vision-language connector, achieving strong performance. These efforts demonstrate the potential of Multi-modal Large Language Models (MLLMs) for complex multi-modal tasks.

**Using MLLMs for Visual Recognition.** Inspired by that LLM have unified various NLP tasks into a generation problem in one architecture, recent MLLM works manifest to solve traditional visual recognition tasks in a unified MLLM architecture. Object detection, a key visual recognition task, poses a challenge in expressing positional information in language within MLLM frameworks. Some literature [6, 20, 21, 2] convert the bounding box into natural language format and directly generate them in text response. However, representing bounding boxes in text form may not be optimal since the bounding box coordinates are numerical data and are typically predicted by regression. Some works [22, 23] use output embeddings of LLM as the understanding pivot to call an object detector for detection. However, the LLM can not benefit from the object detection results from the detector to enhance its own understanding. A more challenging scenario is the segmentation task, in which the ground-truth can be in random shape and seems indescribable via language. To tackle this problem, VisionLLM [24] represents the segmentation mask in text format by representing the mask by the coordinates of the mask polygon. Sphinx [21] and LLaVA-Plus [25] utilize an offline model SAM [26] for segmentation. They first generate the bounding box of the object to segment using MLLM, then they prompt SAM to generate the segmentation mask. LiSA [27] integrates SAM into MLLM for joint training. They introduce a special "<SEG>" token to predict the segmentation mask. However, a single "<SEG>" token cannot differentiate between multiple instances, limiting it to outputting only one segmentation mask.

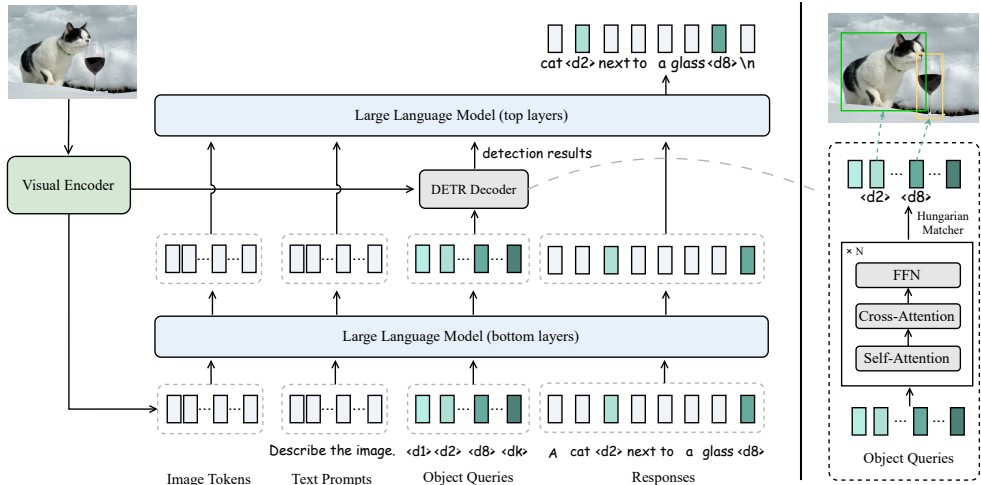

Figure 2: The overall training process of Octopus. We omit the visual backbone to highlight the LLM head. Octopus inputs multiple object queries into the LLM head, in addition to the image tokens and text prompt tokens,. After passing through several bottom LLM layers, the object queries are fed into a DETR decoder for parallel recognition. The recognition is not the canonical close-set detection, but can be class-agnostic detection, visual grounding, referring segmentation, *etc.*, depending on the user prompt (as shown in Fig.3). Afterwards, these object queries, coupled with the image token and text tokens, are then sent into the upper LLM layers for sequential understanding. When the users ask for spotting the mentioned objects (*e.g.*, visual grounding), Octopus finds the object query that detects each object (*e.g.*, the 2nd object query detects the cat) and points out this object query via a corresponding index token "<d2>". The structure of Octopus and its training details are elaborated in Section 3.2 and Section 3.3.

## 3 Approach

### 3.1 Preliminaries

**Sequential inference in prior MLLMs.** Prior MLLMs [1, 2, 21] solve traditional visual recognition tasks (e.g., object detection, instance segmentation) in a sequential generation paradigm. Specifically, they represent the outputs of visual recognition in a natural language format. For example, the bounding boxes are denoted by the coordinates of their corners: "$[x_{min}, y_{min}, x_{max}, y_{max}]$", and segmentation masks are represented by coordinates of points in polygon mask: "$[x_1, y_1, x_2, y_2, \dots]$". Hereby, each numerical value is expressed as multiple text tokens. Sequentially generating all these position tokens can be quite time-consuming, *e.g.*, 25 tokens for a bounding box "[0.152,0.475,0.451,0.692]". Moreover, generating each token at its core, is based on classification. In contrast, coordinates are inherently numerical and are typically predicted through regression. This discrepancy suggests that sequential generation may not be the optimal approach for visual recognition.

**A revisit to DEtection TRansformer (DETR).** Octopus uses a light-weight DETR decoder to cooperate with the bottom LLM layers for parallel visual recognition, as illustrated in Fig. 2. We first give a brief revisit to DETR below.

DETR [9] is an end-to-end object detection approach based on transformer. A DETR model consists of a visual backbone, an encoder, and a transformer decoder. The backbone and the encoder transforms an input image into image feature $\mathbf{F}$. Afterwards, the DETR decoder employs a set of parallel object queries $\mathbf{Q} = \{q_1, q_2, \dots, q_k\}$ to absorb image features through stacked cross-attention layers, which is formulated as:

$$\bar{\mathbf{Q}} = \mathtt{Decoder}(\mathbf{F}, \mathbf{Q}), \tag{1}$$

in which $\bar{\mathbf{Q}}$ is the output state of object queries. Finally, the DETR decoder append class and box predictors upon the object queries to predict their category and bounding box, respectively:

$$\mathbf{B} = \mathtt{box}(\bar{\mathbf{Q}}), \quad \mathbf{S} = \mathtt{cls}(\bar{\mathbf{Q}}), \tag{2}$$

User: Describe the image with positions.
Octopus: Two girls <d50> <d26> sit on the grass <d19> beside a lake <d18>.

User: Where is the white-pink umbrella?
Octopus: <d9>

User: What fruits are in the plate?
Octopus: It contains apples and rambutan.

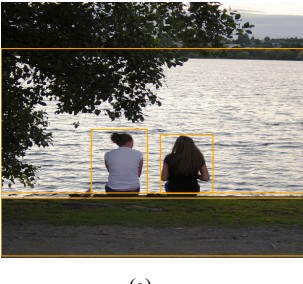 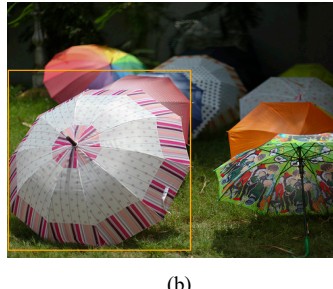 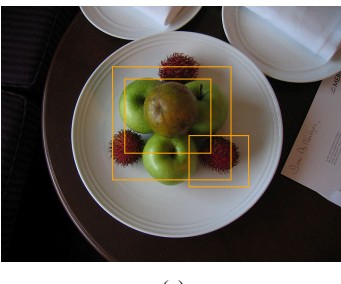

(a)                          (b)                          (c)

Figure 3: **Visualization the Results.** We visualize the detection results and the LLM output from Octopus on three tasks: (a) Spotting Caption, (b) Referring Expression Comprehension (REC), and (c) Visual Dialogue. In the Spotting Caption task, DETR identifies all foreground objects (highlighted in blue), while in REC, DETR locates only the object-of-interest (highlighted in green) as dictated by the prompts. DETR seamlessly transitions between these two modes based on user input. In the case of Visual Dialogue, since the users do NOT ask the MLLM to output object positions, the LLM output does not contain box information, correspondingly. However, Octopus can still localize the objects from the intermediate recognition results (we visualize the detection results with a classification score greater than $0.5$).

in which $\mathbf{B}$ is the bounding boxes and $\mathbf{S}$ is the classification scores. In Octopus we employ DETR as a class-agnostic detector, $\mathbf{S} \in \mathbb{R}^1$ denotes whether an object is foreground or background. Consequently, DETR is independent of the number of classes in the training dataset.

We choose DETR decoder to implement visual recognition for the following reasons: 1) The object queries allow for parallel recognition. 2) The versatility of DETR offers the potential to handle more visual recognition tasks. In addition to object detection, DETR can be adapted for many other recognition tasks, *e.g.*, segmentation [28, 29, 30], pose estimation [31, 32, 33], and object tracking [34, 35, 36]. 3) The object queries can absorb user prompts through attention layers. This allows Octopus to recognize random objects described in natural language, as will be detailed in the following section.

## 3.2 The structure of Octopus

The overall architecture of Octopus is shown in Fig. 2. The image features $\mathbf{F}$ extracted the visual encoder is projected into image tokens $\mathbf{V} = \{v_1, v_2, \ldots, v_m\}$. The text prompts are tokenized and encoded into text tokens $\mathbf{T} = \{t_1, t_2, \ldots, t_n\}$. Based on this standard MLLM structure, Octopus additionally employs $k$ object queries $\mathbf{Q} = \{q_1, q_2, \ldots, q_k\}$. Each $q_i \in \mathbb{R}^D$ has the same dimension $D$ as the image and text tokens. The object queries are placed after image tokens and prompt text tokens. Consequently, $\mathbf{V}, \mathbf{T}$ and $\mathbf{Q}$ are concatenated and jointly fed into the LLM head, which facilitates *parallel recognition* and then *sequential understanding* as below.

**Parallel recognition.** In the bottom LLM layers, the object queries $\mathbf{Q}$ interact with the image tokens $\mathbf{V}$ and text tokens $\mathbf{T}$ through attention. This interaction aligns the object queries with the hidden states of user prompts and image tokens. In benefit of the language knowledge that is embedded in the LLM layers, the object queries thus absorb information from the user prompts and get an understanding of the user interest. It makes the recognition adaptive to user prompts and is crucial for visual grounding tasks, *e.g.*, referring expression comprehension (REC). Afterwards, the object queries are fed into a lightweight (4-layers) DETR decoder to detect the objects. Both the $\mathbf{Q}$-$\mathbf{V}$-$\mathbf{T}$ interaction and the object detection DETR decoder are in parallel, yielding the complete recognition for Octopus.

*Discussion*: It's worth noting that the key difference between the integrated DETR decoder in Octopus and traditional DETR trained on closed-set is that in Octopus the DETR decoder makes predictions based on user prompts. The predictions made by Octopus DETR can be class-agnostic detection, visual grounding, referring segmentation, *etc.*, depending on the user prompt. Fig. 3 shows DETR predicts the objects-of-interest depending on user prompts. For example, *e.g.*, it tries to locate all objects in the image, given the user prompt "Please detect all objects in the image". Given another

user prompt "Please spot the black-and-white cat in the image", DETR focuses on identifying the specified cat rather than other objects.

**Sequential understanding.** The recognition results, *i.e.*, the object queries output from the DETR decoder, coupled with the hidden states of $\mathbf{V}$ and $\mathbf{T}$, are relayed into the following LLM layers. These recognition results will be absorbed into the final output tokens which form the model response, *e.g.*, image captions or visual question answers, in a sequential manner.

In visual grounding tasks, the model response is expected to contain bounding boxes for the mentioned objects. In the previous MLLMs, the bounding box is represented in natural language form, which takes multiple tokens and is time-consuming to generate (Section 3.1). In contrast, Octopus does not generate the text of the bounding box in a sequential manner, but instead, it selects the detection boxes predicted by the DETR. For example, in Fig. 2, the 2nd object query from the DETR decoder detects the cat. In the LLM output, Octopus generates a special token "<d2>" which indexes the 2nd object query after the "cat" token. We name the token that indexes an object query as **index token**. Consequently, Octopus becomes aware of the position of the cat by selecting the bounding box corresponding to the predicted index token. How Octopus learns to predict the index tokens is elaborated in the following Section 3.3.

### 3.3  Training Octopus

Training Octopus involves supervision of two components: the DETR output and the LLM output. These components are trained jointly, meaning that both the DETR decoder and the LLM are optimized together.

**Supervision on the DETR output.** Given the detection predictions $\mathbf{Y} = \{(s_1, \mathbf{b}_1), \ldots, (s_k, \mathbf{b}_k)\}$ and the ground-truth objects $\bar{\mathbf{Y}} = \{(1, \bar{\mathbf{b}}_1), \ldots, (1, \bar{\mathbf{b}}_N)\}$, DETR uses the Hungarian algorithm to find the optimal assignment $\sigma(\cdot)$, where each ground-truth object is assigned to its best-matched prediction. Here, $(s_i, \mathbf{b}_i)$ indicates the predicted classification scores and bounding boxes from query $q_i$. All objects in the targets are treated as foreground objects, and thus, a binary classifier is used to predict whether an object query is foreground or background (non-object). The classification loss $\ell_{\text{cls}}(\cdot)$ is computed using binary cross-entropy, and the box regression loss $\ell_{\text{box}}(\cdot)$ is computed using L1 box distance and GIoU loss:

$$\mathcal{L}_{\text{DETR}} = \sum_{n=1}^{N} (\ell_{\text{cls}}(s_{\sigma(n)}, \bar{s}_n) + \ell_{\text{box}}(\mathbf{b}_{\sigma(n)}, \bar{\mathbf{b}}_n)), \tag{3}$$

**Supervision on the LLM output.** The LLM output is supervised through the next-token-prediction manner. However, supervising the object position differs significantly. We recall that for spotting objects, the LLM head does not generate the bounding boxes through text, but predicts an index token that points to the corresponding detection result. Correspondingly, during training, Octopus is trained to predict the index token following each mentioned object. The ground-truth index token is not fixed, but dynamically determined on-the-fly in each training iteration.

Identifying the index token requires matching the ground-truth object to its nearest object query. To this end, we get the predicted bounding box $\mathbf{b}$ (the subscript is omitted) of all object queries, and then find the nearest query for the ground-truth object at $\mathbf{b}$ by:

$$C_{\text{loc}} = \left\| \bar{\mathbf{b}} - \mathbf{b} \right\| + \left( 1 - \text{GIoU}(\bar{\mathbf{b}}, \mathbf{b}) \right), \tag{4}$$

We replace the bounding boxes in the response with the index tokens to the matched object queries. This process introduces slight overload to the training (increasing $\sim 10\%$ training time). In inference, the index tokens are directly generated in the response. Since we exclusively use index tokens to represent bounding boxes without the need to generate bounding boxes token-by-token, our method is significantly faster than previous MLLMs.

The LLM computes the training loss as the language modeling loss using next-token prediction, the same as prior MLLMs.

$$\mathcal{L}_{\text{LM}} = -\sum_{i=1}^{K} \log P(y_i | y_{<i}), \tag{5}$$

where $y_i$ represents the target token at position $i$. The overall training objective is the combination of DETR training loss and language modeling loss.

Table 1: **Results on Referring Expression Comprehension benchmarks.** We note that Octopus with resolution 224/336 is up to $5\times/4\times$ faster than a purely sequential counterpart Shikra (resolution 224) and achieves higher accuracy. More details of the inference speed comparison are reported in Sec. 4.5

| Model Type | Method | Res. | RefCOCO | | | RefCOCO+ | | | RefCOCOg | |
|---|---|---|---|---|---|---|---|---|---|---|
| | | | val | test-A | test-B | val | test-A | test-B | val | test |
| Generalists | OFA-L[37] | 480 | 79.96 | 83.67 | 76.39 | 68.29 | 76.00 | 61.75 | 67.57 | 67.58 |
| | VisionLLM[24] | 224 | - | 86.70 | - | - | - | - | - | - |
| | Shikra [6] | 224 | 87.01 | 90.61 | 80.24 | 81.60 | 87.36 | 72.12 | 82.27 | 82.19 |
| | MiniGPT-v2 [38] | 448 | 88.06 | 91.29 | 84.30 | 79.58 | 85.52 | 73.32 | 84.19 | 84.31 |
| | **Octopus** | 224 | **88.77** | **91.93** | 82.28 | **83.05** | **88.87** | **75.12** | 83.11 | **84.78** |
| | **Octopus** | 336 | **89.02** | **92.63** | 83.42 | **83.55** | **89.40** | **76.02** | **84.25** | **86.19** |
| Specialists | G-DINO-L [39] | 512 | 90.56 | 93.19 | 88.24 | 82.75 | 88.95 | 75.92 | 86.13 | 87.02 |
| | UNINEXT-H [40] | 640 | 92.64 | 94.33 | 91.46 | 85.24 | 89.63 | 79.79 | 88.73 | 89.37 |

## 4 Experiments

### 4.1 Settings

**Training details.** We train Octopus via three stages, *i.e.*, stage-1 for pretraining vision-language alignment, stage-2 for pretraining the DETR recognition module, and stage-3 for end-to-end instruction fine-tuning. The details are as below:

**1)** Stage-1 pretrains the vision-language connector for vision-language alignment using LLaVA pretraining data [18]. The visual encoder and LLM are frozen and only the vision-language connector is trained. **2)** Stage-2 pretrains the DETR module on small-scale grounding detection datasets (RefCOCO [41], RefCOCO+ [42], RefCOCOg [42] and Flickr30k [43]) to quickly obtain the recognition ability. We freeze LLM and visual encoder and only train the DETR module in this stage. Stage-2 is mainly for training efficiency, *i.e.*, fast adapting the DETR decoder to the LLM layers. **3)** In stage-3, we finetune the whole LLM head and DETR decoder on LLaVA-Instruct [18], REC data (RefCOCO, RefCOCO+, RefCOCOg, Visual Genome [44]), and Flickr30k end-to-end. Please refer to the Appendix for details of the training datasets.

We adopt AdamW as the optimizer and cosine annealing scheduler. The learning rate is initialized to 1e-4 for stage-1 and stage-2, and 2e-5 for stage-3. The entire training takes about 2 hours for Stage-1 (1 epoch), 4 hours for Stage-2 (2 epochs) and 120 hours for Stage-3 on 8 NVIDIA A100 GPUS.

**Architecture details.** Octopus adopts the ViT pre-trained from CLIP as the visual encoder. All the vision-language connectors are one-layer MLP with random initialized. The LLM head is initialized with Vicuna-7B-v1.5 [45] and the DETR decoder consists of 4 standard DETR decoder layers. We employ 64 object queries and place the DETR decoder after the 16-th LLM layer.

### 4.2 Evaluation on REC datasets

We evaluate Octopus's recognition ability on 3 popular referring expression comprehension datasets, *i.e.*, RefCOCO, RefCOCO+ and RefCOCOg. In Table 1, Octopus achieves the highest accuracy on seven out of eight dataset splits, among the compared generalist MLLMs. For example, Octopus outperforms MiniGPT-v2 [46] by +0.96% on val split and +1.34% on test-A split of RefCOCO dataset. On RefCOCO+, the superiority is even larger, *e.g.*, surpassing MiniGPT-v2 by +3.97% on val split, +3.92% on test-A split and +2.70% on test-B split.

We particularly note the comparison against Shikra, a purely sequential counterpart that adopts the same backbone as our Octopus and shares the same training data. Octopus (resolution 224) is up to $5\times$ faster than Shikra under the same image resolution and consistently achieves higher accuracy, *e.g.*, +1.51% on RefCOCO+ test A. When scaling up the image resolution to 336, Octopus is still $4\times$ faster than Shikra (resolution 224) and further enlarges the accuracy superiority to + 2.04% on RefCOCO+ test A. These observations show that parallel recognition improves both the efficiency and accuracy for recognition.

Table 2: **Results on Visual Question Answering benchmarks.** Note that specialists are fine-tuned on each individual evaluation dataset. We gray out those specialists methods, as well as the fine-tuned results of generalists.

| Model Type | Method | #LLM Params | Res. | VQAv2 | OKVQA | GQA | VizWiz | SciQA |
|---|---|---|---|---|---|---|---|---|
| Generalists | BLIP2 [3] | 11B | 224 | 65.0 | 45.9 | 41.0 | 19.6 | 61.0 |
| | InstructBLIP [4] | 11B | 224 | - | - | 49.2 | 34.5 | 60.5 |
| | Unified-IO$_{XL}$ [47] | 2.7B | 256 | 77.9 | 54.0 | - | - | - |
| | PaLM-E-12B [48] | 12B | 224 | 76.2 | 55.5 | - | - | - |
| | Shikra [6] | 7B | 224 | 77.4 | 47.2 | - | - | - |
| | **Octopus** | 7B | 224 | **78.5** | **56.0** | **62.29** | **45.6** | **65.7** |
| | LLaVA-1.5 [49] | 7B | 336 | 78.5 | - | 62.0 | 50.0 | 66.8 |
| | Qwen-VL-Chat [50] | 7B | 448 | 78.2 | 56.6 | 57.5 | 38.9 | **68.2** |
| | **Octopus** | 7B | 336 | **79.2** | **57.2** | **63.3** | **50.1** | 67.7 |
| Specialists | GIT [51] | 0.7B | 384 | 78.6 | - | - | 68.0 | - |
| | GIT2 [51] | 5.1B | 384 | 81.7 | - | - | 71.0 | - |
| | PaLI-17B [52] | 17B | 580 | 84.3 | 64.5 | - | 71.6 | - |

Table 3: **Results on popular VL benchmarks.** MMB is MMBench [53], LLaVA$^W$ is LLaVA-Bench (In-the-Wild) [18] and MM-V is MM-Vet Benchmark[54]. POPE [55] is reported on the average F1 score of three splits (Adersarial, Popular and Random).

| Method | #LLM Params | Resolution | MMB | LLaVA$^W$ | SEED | MM-V | POPE |
|---|---|---|---|---|---|---|---|
| BLIP-2[3] | 13B | 224 | - | 38.1 | 46.4 | 22.4 | - |
| InstructBLIP [4] | 7B | 224 | 36.0 | 60.9 | 53.4 | 26.2 | - |
| InstructBLIP [4] | 13B | 224 | - | 58.2 | - | 25.6 | 78.9 |
| IDEFICS [16] | 7B | 480 | 48.2 | - | - | - | - |
| IDEFICS [16] | 65B | 480 | 54.5 | - | - | - | - |
| LLaVA-1.5 [49] | 7B | 336 | 64.3 | 65.4 | 58.6 | 31.1 | 84.2 |
| QwenVL-Chat [50] | 7B | 448 | 60.6 | - | 58.2 | - | - |
| Shikra [6] | 7B | 224 | 58.8 | - | - | - | 83.9 |
| **Octopus** | 7B | 224 | **66.2** | 63.9 | **58.6** | **32.1** | **84.8** |

## 4.3 Evaluation on VQA datasets

General visual question answering (VQA) is a widely employed task for MLLMs. We compare Octopus on 5 VQA datasets against multiple competing MLLMs in Table 2. It is observed that Octopus achieves competitive results under both 224 and 336 resolution settings. For instance, at a 224 image resolution setting, Octopus outperforms Shikra [6] by +1.10Moreover, it surpasses InstructBLIP [4] by +13.09%, +11.1%, and +5.2% on GQA, VizWiz, and SciQA, respectively.

We note that on these datasets, MLLMs are not asked to provide position of the mentioned objects, but Octopus still gains benefit from parallel visual recognition. This suggests that the recognition effectively supports the subsequent understanding, validating the benefit of "parallel recognition → sequential understanding" hierarchy.

## 4.4 Evaluation on recent MLLM benchmarks

We report the performance on the recent popular MLLM benchmarks in Table 3. It is observed that Octopus performs favably against purely sequential counterparts on the five benchmarks. For example, Octopus is higher than Shikra by +7.4% on MMBench and +0.9% on POPE, respectively.

Table 4: **Comparison of inference speed.** We compared the inference speed of our method with the baseline method Shikra [6] on two benchmarks. The reported inference times (in seconds) are averaged for one record across the datasets. RefCOCO contains exactly 1 bounding box in outputs, and Flickr30k contains on average 4 boxes in outputs. Our method is much faster ($\sim 5\times$) than the baseline in both resolutions.

| Method | #LLM Params | Resolution | RefCOCO | | Flickr30k | |
|---|---|---|---|---|---|---|
| | | | infer time ↓ | FPS ↑ | infer time ↓ | FPS ↑ |
| Baseline | 7B | 224 | 0.89 | 1.12 | 3.80 | 0.26 |
| **Octopus (ours)** | 7B | 224 | 0.17 | 5.88 | 0.82 | 1.21 |
| Baseline | 7B | 336 | 1.16 | 0.86 | 6.08 | 0.16 |
| **Octopus (ours)** | 7B | 336 | 0.27 | 3.70 | 1.49 | 0.67 |

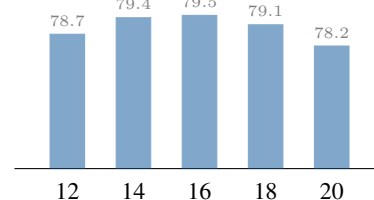 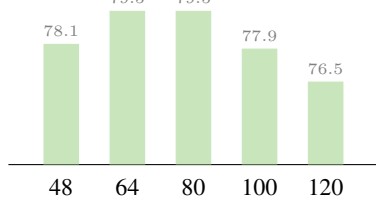

Figure 4: **Influence of the recognition layers and number of object queries.** We evaluate our method on RefCOCOg val benchmark. We did not use Visual Genome data [56] in training for efficiency. (a) Influence of the number of LLM layers employed for recognition. (b) Influence of the number of object queries.

Octopus also outperforms InstructBLIP by +3.0% on LLaVA-Bench, +5.2% on SEED and +6.5% on MM-Vet, respectively.

## 4.5 Ablation Study

**The number of LLM layers employed for recognition.** We study the optimal layer for positioning DETR within LLM. Fig. 4 (a) illustrates our method's performance on the RefCOCOg val benchmark when placing DETR after different LLM layers. We find that the optimal placement for DETR is after the 16th LLM layer, which is the middle layer of the LLM (consisting of 32 layers in total). We infer that positioning DETR on a lower layer might limit its comprehension of the instruction prompt, while placing it at a higher layer harms the ability of LLM to infer the correct index tokens for the mentioned objects.

**Number of object queries.** We compare the impact of different object query number on performance in Fig. 4 (b). We evaluate Octopus on the val split of RefCOCOg benchmark and observe that 64 and 80 object query achieved the best performance in our setup. We infer that using insufficient object queries will hamper the recognition capabilities of the bottom LLM layers. On the other hand, although an excess of object queries can enhance the LLM's perception ability for foreground objects, it also increases the complexity for the LLM to select the correct object queries in the final output.

**Inference speed.** We compare the inference speed of our method against the traditional MLLM on the grounding detection data. As shown in Table 4, our method is much faster ($5\times$) on all benchmarks. This is attributed to that our method does not require to generate bounding boxes in discrete tokens. For each bounding box, we only need to generate an index token to the corresponding detection results predicted by the DETR module. It saves 24 tokens in generating one box, which is more significant in scenarios where a larger number of objects exist.

**Qualitative analysis.** We visualize the recognition results predicted by DETR for non-grounding data, such as VQA and instruction dialogue, in which no bounding boxes are provided and generated. As shown in Fig. 3 (c), DETR still locates the mentioned objects in the user prompt even if it is not in grounding data format. The detection results are helpful for VQA and some tasks, where the MLLM needs to locate the mentioned objects in the image and answer specific questions.

# 5 Limitation and Conclusion

**Limitations.** Due to limited GPU resources, we have not been able to explore how Octopus would perform when scaling up on larger LLM and more open-ended instruction-tuning data. Moreover, DETR is applicable to many recognition tasks beyond detection. We leave it as future works.

**Conclusion.** We propose Octopus, a novel MLLM framework that disrupts the purely sequential inference paradigm for LLM head. Octopus perform parallel recognition through the lower LLM layers and a lightweight DETR decoder, and then passes the recognition results to the upper LLM layers for further understanding. Consequently, it reformulates the LLM head into a "parallel recognition → sequential understanding" hierarchy. Empirical results show Octopus improves accuracy over a range of MLLM tasks and significantly enhances inference efficiency when the task include recognition objectives.

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

# A  Comparison between the intermediate and final recognition results

In Octopus, there are two components that can give recognition results, *i.e.*, the intermediate outputs from the DETR and the final LLM outputs. The intermediate recognition results are parallel and redundant, *i.e.*, predicting multiple boxes for each object of interest. Based on these parallel recognition results, the final LLM outputs choose a single one for each object via the indexing tokens.

We compare these two different results for recognition objectives in Table 5. Since the intermediate recognition results (DETR results) have multiple predictions, we evaluate the rank-$k$ accuracy, which indicates the recall of the ground-truth. We draw two observations as below:

First, the final LLM outputs yield higher accuracy than the rank-1 DETR results, *e.g.*, +1.18% and + 2.12% under the 224 and 336 resolution settings, respectively. It shows that the LLM top layers further improves recognition accuracy after understanding all the recognition results.

Second, the redundant DETR results can achieve high recall of the object-of-interest, *e.g.*, 94.56% rank-5 accuracy under the 224 resolution. Since these intermediate detection results are relayed into the LLM top layers, we infer the high recall is an important reason for the LLM to improve recognition accuracy by selecting one result for each object-of-interest.

In addition to the benefit of improving recognition accuracy, the understanding part of Octopus has another important role: it organizes multiple recognition results into a complete description or response, *e.g.*, in the spotting caption task (Fig. 5 in the main part).

Table 5: **Comparison of Detection Performance.**

| Method | Resolution | Octopus Accuracy | DETR results | | | | |
|--------|------------|---------|------|------|------|------|------|
| | | | R@1 | R@2 | R@3 | R@4 | R@5 |
| Octopus | 224 | 83.11 | 81.93 | 90.54 | 92.85 | 93.89 | 94.56 |
| Octopus | 336 | 84.25 | 82.13 | 92.79 | 94.58 | 95.36 | 95.95 |

# B  Application on referring segmentation

In addition to detection, Octopus is potential to acquire a broader range of recognition abilities, taking advantage of the versatility of the DETR mechanism. Hereby, we endow Octopus with the referring segmentation ability by modifying its object queries into "mask" queries, a common

| Where is the bus in center? | Where is the chocolate cake? | Where is the front-most bus? |
|---|---|---|
| Where is the left yogurt? | Where is the right sandwich? | Where is the right bowl? |

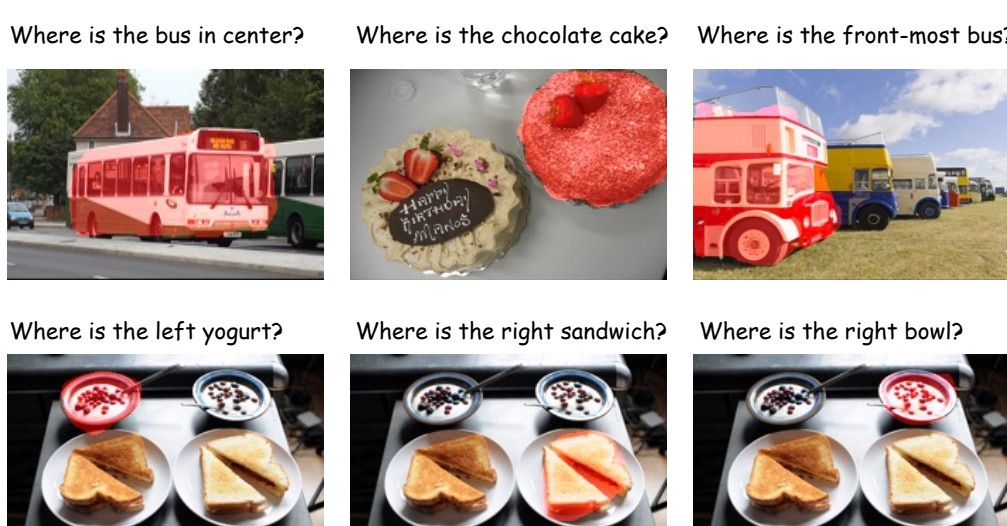

Figure 5: **Visualization of the referring segmentation results. Octopus directly makes pixel-wise predictions rather than predicting the vertexes of the segmentation mask.**

application of DETRs. We add an additional mask head on top of the DETR decoder to predict the segmentation mask for each mask query. Under this setting, we train Octopus on the training datasets of referring segmentation datasets (RefCOCO, RefCOCO+, RefCOCOg). Visualization in Fig. 5 shows that Octopus gains referring segmentation ability. The ability of directly predicting pixel-wise segmentation results rather than vertexes of segmentation masks is valuable for MLLMs.

Table 6: **Referring segmentation results on RefCOCO.**

| Method | Resolution | val | testA | testB |
|--------|-----------|-----|-------|-------|
| Octopus | 224 | 63.6 | 66.6 | 61.3 |

We quantitatively evaluate the referring segmentation performance on the RefCOCO benchmark using cIoU. Table. 6 show that our model achieves reasonable results. Due to time limit, we only implement Octopus using the low-resolution CLIP-ViT features ($224 \times 224$) for this experiment. The input size is very small for segmentation task and is an important reason that limits our performance (*e.g.*, 63.6 cIoU on RefCOCO validation set). We note a recent state-of-the-art method LISA achieves 74.6. LISA uses an external strong segmentation model, SAM [26], that is pretrained on SA-1B datasets and uses large input size. We conjecture that enlarging the input size and adding training data will bring further improvement to Octopus, as well.

To sum up, by incorporating the bottom layers of LLM head with the DETR query mechanism, Octopus is potential to acquire various recognition abilities. We will explore more forms of recognition abilities for Octopus.

