# OpenReview forum: "Octopus: A Multi-modal LLM with Parallel Recognition and Sequential Understanding"
_NeurIPS.cc/2024/Conference — NeurIPS 2024 poster_

### Official Review · Reviewer_u3Ag · 2024-06-17

**Soundness:** 2
**Presentation:** 2
**Contribution:** 2
**Rating:** 5
**Confidence:** 5

**Summary:**

The paper presents Octopus, a combined MLLM for joint localization and visual chat tasks. The authors insert the DETR decoder into the LLM to perform visual grounding to replace LLM’s outputs for more precise localization. The bottom LLM layers are utilized for parallel recognition and the recognition results are relayed into the top LLM layers for sequential understanding. Extensive results on various MLLM tasks show the effectiveness of Octopus.

**Strengths:**

1, The overall writing is easy to understand and follow.

2, The proposed model, Octopus, can perform both localization tasks and visual chat tasks in one shot. Moreover, it runs faster than previous works.

3, The performance looks good compared with several previous MLLM, such as MiniGPT-V2 and Shika.

**Weaknesses:**

1, The motivation of combining DETR with MLLM is not new. Several previous works have explored the nearly same design. In particular, LLaVA-Grounding have the same design by introducing the Detr decoder into the output of LLM. However, as shown in Fig.4 (a), taking the intermediate LLM layer as the input of DETR, the effects are not that significant than previous design in LLaVA-Grounding.

2, The technical novelty is limited. The meta-architecture is the same as previous works.
[2] is on arXiv even one year ago. However, the authors adopt the nearly same idea but without a citation. This is disrespectful for the community.

[1], LLaVA-Grounding: Grounded Visual Chat with Large Multimodal Models, arxiv-2023.

[2], Contextual Object Detection with Multimodal Large Language Models, arxiv-2023.


3, Several ablation studies are missing.   For example, the effect of pre-training data and the effect of transformer layers in DETR.

4, No failure case analysis and no improvement analysis, compared with late fusion decoder.

5, What about using LLM to predict the box in the end?

**Questions:**

See the weakness.

---

> ### Author Rebuttal · Authors · 2024-08-07
>
> **Q1**: The motivation of combining DETR with MLLM is not new. Several previous works have explored the nearly same design. In particular, LLaVA-Grounding have the same design by introducing the Detr decoder into the output of LLM.
> **A1**: Our motivation is not combining DETR with MLLM.
> Our motivation (and key contribution) is integrating the parallel recognition and sequential understanding into one LLM, and organizing them into a “parallel recognition → sequential understanding” hierarchy. This is fundamentally different and even opposite to previous "MLLM + DETR'' methods because they all follow the "understanding → recognition'' regime, i.e., first concept understanding and then recognition.
>
> Our novel hierarchical MLLM framework has several advantages, compared to previous "DETR + MLLM'' methods, including:
> - The parallel recognition results can further augment the sequential understanding ability of current MLLMs. In contrast, in previous "MLLM + DETR'' methods, the recognition does not benefit understanding.
> - Our DETR head can be used separately for recognition tasks, resulting in high efficiency. In contrast, previous "MLLM + DETR'' methods is more computational expensive because they require full inference over the entire LLM.
>
> ---
>
> **Q2**: The technical novelty is limited. The meta-architecture is the same as previous works. [2] is on arXiv even one year ago. However, the authors adopt the nearly same idea but without a citation. This is disrespectful for the community.
> **A2**: We respectfully disagree. Though our method shares some similarity ("MLLM + DETR''), the architecture and underlying mechanism are fundamentally different:
> - Architecture: as explained in the Q1-Answer, our Octopus is a novel "parallel recognition → sequential understanding'' MLLM framework, while [2] adopts the popular "understanding pivot → recognition head'' regime.
> - Mechanism: [2], as well as some recent methods [1,3], employs an MLLM as the understanding pivot to call a recognition head, so that the recognition head can receive and understand free-style human instructions. The LLM benefits the recognition head. In contrast, our Octopus uses bottom LLM layers to assist the recognition and in turn uses the recognition results to prompt the following understanding. The LLM and the recognition mutually benefit each other.
> We will add the above comparison to the manuscript.
>
> ---
>
> **Q3**: Several ablation studies are missing. For example, the effect of pre-training data and the effect of transformer layers in DETR.
> **A3**: Thanks. The pretraining data consists of REC and Flickr30k and the ablation is shown in the table below. During rebuttal, we further increased the pretraining data by adding Visual Genome (3rd row) and observed another round of improvement.
>
> | pre-training data | accuracy |
> | ----------------- | -------- |
> | REC               | 79.2     |
> | +Flickr30k        | 79.5     |
> | +Visual Genome    | 79.9     |
>
> The impact of the number of transformer layers in DETR is presented in the table below. We observe that increasing the number of transformer layers results in a slight improvement in accuracy from 4 to 6 layers, reaching an optimal performance at 6 layers. Considering training efficiency, we use 4 layers in our experiments.
>
> | \#DETR layers | accuracy |
> | ------------- | -------- |
> | 4             | 79.5     |
> | 5             | 79.7     |
> | 6             | 79.8     |
> | 7             | 79.7     |
>
> ---
>
> **Q4**: No failure case analysis and no improvement analysis, compared with late fusion decoder.
> **A4**: We guess the “late fusion decoder'' you mentioned is the ”understanding $\rightarrow$ recognition'' methods [1,2,3]. Compared with late fusion decoder, our method used relatively fewer LLM layers (16 out of 32 layers) for informing the DETR head with the object-of-interest. It is possible that in some extreme cases where the object-of-interest is very obscure, our model do not capture the user intention and missed the object.
> The advantage of our approach, on the other side, is that the recognition is much more efficient for recognition (REC, in particular). More importantly, our recognition results will be injected into the following LLM layers to assist the understanding. In contrast, the late fusion mode does not benefit understanding.
>
> ---
>
> **Q5**: What about using LLM to predict the box in the end?
> **A5**: Directly using LLM to predict the position is the "sequential recognition'' manner introduced in Line 29 to 32 in the manuscript and is inferior to ours. One important motivation of this paper is that the sequential recognition manner has low efficiency. Table 4 in the manuscript shows that our method significantly improves the efficiency by about 4 $\times$ faster. Table 1 in the manuscript shows that our method has higher accuracy (e.g., 2.01 on RefCOCO) in comparison, as well.
> Reviewer Up36 kindly reminds there is another manner, i.e., using localization tokens, as in Kosmos-2. This manner has relatively lower REC accuracy (-5.9 acc. on RefCOCO test set) and is inferior to our method. Please refer to Reviewer Up36 Q2 for details.

---

> > ### Comment · Reviewer_u3Ag · 2024-08-13
> > **Boadline accept at highest score**
> >
> > Thanks for the rebuttal and response. Several concerns are solved. I raise my score as borderline accept.
> >
> > 1, “Though our method shares some similarity ("MLLM + DETR'')” in the rebuttal, the authors should cite and compare them fairly since they are very close. In addition, these works are proposed nearly one year ago. This is extremely disrespectful to community. In rebuttal, authors also do not give detailed comparison to [1].
> >
> > 2, The experiment results are still not convincing and fair, as also pointed out by Reviewer Up36.
> >
> > Based on these, despite other reviewers give weak accept, I only can give board line accept as highest score.

---

> > > ### Author Response · Authors · 2024-08-13
> > >
> > > Thank you for your valuable feedback and for raising your rating to a positive score.  We sincerely apologize for overlooking the "MLLM + DETR" works in our initial submission. We appreciate your reminder and recognize the importance of discussing these works. The comparison during the rebuttal clarifies our contributions and highlights the novelty of our approach. We will ensure that these works are properly cited and add the above comparison to our manuscript. We are committed to refining our manuscript to further enhance its quality.

---

> ### Comment · Area_Chair_PhW3 · 2024-08-13
>
> Dear Reviewer,
>
> Thanks for reviewing this paper! The authors have provided rebuttal to address your concerns. Could you have a look and let the authors know if you have further questions?
>
> Thanks,
> Your AC

---

### Official Review · Reviewer_Up36 · 2024-07-12

**Soundness:** 3
**Presentation:** 3
**Contribution:** 2
**Rating:** 5
**Confidence:** 5

**Summary:**

This paper introduces Octopus, a multimodal LLM that employs parallel recognition and sequential understanding. Inspired by the human brain, the authors aim to address the efficiency issues inherent in existing methods by proposing this novel approach. The paper makes three key contributions: (1) Identification of an efficiency issue with the sequential paradigm, (2) Proposal of the Octopus framework to enable parallel recognition and sequential understanding, and (3) Demonstration of superior performance and inference efficiency compared to the sequential approach.

**Strengths:**

- This paper is well-written and easy to understand. The proposed Octopus framework is novel and intriguing.
- The experimental results partially support the authors' claims.
- The proposed framework not only performs object localization but also generalizes well to pixel-wise recognition tasks such as segmentation.

**Weaknesses:**

- The biggest problem is that the experimental results and comparisons in this paper are not comprehensive and sometimes unfair to some baseline methods. Octopus is built upon LLaVA and leverages additional data beyond LLaVA's dataset. Therefore, the results in Table 2 cannot conclusively demonstrate that Octopus benefits from parallel visual recognition (as stated in Line 251). It would be better to compare Octopus and LLaVA using the exact same training data. Moreover, in Table 3, some methods from Table 2, such as LLaVA and Qwen-VL, are not included. If the result comparisons are cherry-picked, the conclusions are unreliable, and the performance advantages are not significant.
- Previous MLLMs with grounding capacity can be grouped into two categories: (1) generating box coordinates as natural language (like Shikra and Qwen-VL), and (2) introducing localization tokens (such as PaLI and Kosmos-2). This paper addresses the efficiency issue of the former methods (coordinates as language) but ignores the latter. What are the performance and efficiency advantages when comparing Octopus with the latter (introducing new localization tokens)?
- In Line 200, it is stated that after bipartite matching, the response is modified by replacing the bounding boxes with the index tokens. Does this mean that for one sample (image, instruction, and response), it requires dual forward passes (first image + instruction + queries, then image + instruction + queries + modified response)? As the authors mentioned in Line 201, Octopus introduces non-negligible training overhead.
- Considering the unfair comparisons in Tables 2 and 3, and given the slight performance advantages in Table 1, the advantage of "Parallel Recognition" does not seem significant.

**Questions:**

- How are the newly introduced object queries initialized? Does this affect the final performance?
- Would it be better to use a pre-trained DETR decoder from off-the-shelf publicly available checkpoints?

**Limitations:**

Generally speaking, the proposed method is interesting and indeed more efficient than generating bounding box strings sequentially. As such, I am inclined to assign a borderline score (slightly leaning towards acceptance) at this stage. However, this paper also suffers from several issues that could lead to a rejection in the final decision.

---

> ### Author Rebuttal · Authors · 2024-08-07
>
> **Q1**: The comparison in this paper are not comprehensive and sometimes unfair to some baseline methods. It would be better to compare Octopus and LLaVA using the exact same training data. Moreover, in Table 3, some methods from Table 2, such as LLaVA and Qwen-VL, are not included.
> **A1**: Thank you for your advice. We have trained LLaVA with the same source of training data as Octopus (by adding the grounding data). The results in below Table show Octopus surpasses LLaVA with same training data.
>
> | Method  | \#LLM Params | Res. | VQAv2 | GQA  | VizWiz | SciQA |
> | ------- | ------------ | ---- | ----- | ---- | ------ | ----- |
> | LLaVA   | 7B           | 336  | 78.8  | 62.5 | 48.8   | 65.9  |
> | Octopus | 7B           | 336  | 79.2  | 63.3 | 50.1   | 67.7  |
>
> We will add the results of Qwen-VL and LLaVA into Table 3 in the manuscript. As shown in the below table, our method surpasses LLaVA and QwenVL-Chat on MMB, SEED, MM-V, and POPE benchmarks, but is lower than LLaVA on the LLaVA$^W$ benchmark.
>
> | Method      | \#LLM Params | Resolution | MMB      | LLaVA$^W$ | SEED     | MM-V     | POPE     |
> | ----------- | ------------ | ---------- | -------- | --------- | -------- | -------- | -------- |
> | LLaVA       | 7B           | 336        | 64.3     | **65.4**  | 58.6     | 31.1     | 84.2     |
> | QwenVL-Chat | 7B           | 448        | 60.6     | -         | 58.2     | -        | -        |
> | Octopus     | 7B           | 224        | **66.2** | 63.9      | **58.6** | **32.1** | **84.8** |
>
> ---
>
> **Q2**: Some MLLMs, such as PaLI and Kosmos-2 introdices localization tokens to fullfill grounding ability. What are the performance and efficiency advantages when comparing Octopus with these methods (introducing new localization tokens)?
> **A2**: Thank you for your advice. We compare using localization tokens and our Octopus in the table below. It is observed that our Octopus achieves higher accuracy (e.g., + 5.9 test acc.) and is slightly faster. The results are reasonable: the localization tokens have coarse spatial resolution (7 $\times$ 7 pixels per token) and is thus inferior for REC. We used the Kosmos-2 code for implementing the localization code and adopted the same backbone (i.e., LLaVA-1.5) plus the same REC training data for a fair comparison. We did not use PaLI because PaLI has not released its code yet.
>
> | Method   | Resolution | infer time | fps  | val  | test |
> | -------- | ---------- | ---------- | ---- | ---- | ---- |
> | Kosmos-2 | 224        | 0.23       | 5.34 | 73.2 | 74.3 |
> | Octopus  | 224        | 0.17       | 5.88 | 79.5 | 80.2 |
>
> ---
>
> **Q3**: In Line 200, it is stated that after bipartite matching, the response is modified by replacing the bounding boxes with the index tokens. Does this mean that for one sample (image, instruction, and response), it requires dual forward passes (first image + instruction + queries, then image + instruction + queries + modified response)? As the authors mentioned in Line 201, Octopus introduces non-negligible training overhead.
> **A3**: Yes, our model requires two forward passes to generate the correct response. However, the training overhead is relatively low (only \~10\%) because the first pass requires no gradients computation and is efficient.
>
> ---
>
> **Q4**: Considering the unfair comparisons in Tables 2 and 3, and given the slight performance advantages in Table 1, the advantage of "Parallel Recognition" does not seem significant.
> **A4**:
> The advantage of "parallel recognition" on REC is non-trivial, regarding both efficiency (about $4\times$ faster in Table 4) and accuracy (+1.76 over the baseline on Refcoco val). Please kindly note that we have already made the comparison fair by using the same training data during rebuttal.
>
> As for understanding task (Table 2 and 3), the superiority does seem relatively small. This is because many understanding tasks are not closely related to recognition. During rebuttal, we add another experiment by selecting closely-related sub-tasks and observe larger improvement (+1.7 VQA v2). Please refer to R1-Q1 for details.
>
>
> ---
>
> **Q5**: How are the newly introduced object queries initialized? Does this affect the final performance?
> **A5**: We guess your question is about the training initialization for the object queries. At the start of training, we use the standard Gaussian initialization (the default PyTorch initialization) for these object queries (as well as for all the other vocabulary embeddings). After training convergence, all the object queries use the learned embeddings for each inference.
>
> ---
>
> **Q6**: Would it be better to use a pre-trained DETR decoder from off-the-shelf publicly available checkpoints?
> **A6**: Thanks. Utilizing a pre-trained DETR decoder can benefit training, as validated by our experiments in Stage-2 (L219). A pre-trained decoder helps in quickly adapting the DETR decoder to the LLM layers. If we do not use DETR pretraining, it takes a longer period (1.5$\times$) to achieve similar results.

---

> ### Comment · Area_Chair_PhW3 · 2024-08-13
>
> Dear Reviewer,
>
> Thanks for reviewing this paper! The authors have provided rebuttal to address your concerns. Could you have a look and let the authors know if you have further questions?
>
> Thanks,
> Your AC

---

### Official Review · Reviewer_aS3o · 2024-07-13

**Soundness:** 3
**Presentation:** 3
**Contribution:** 2
**Rating:** 6
**Confidence:** 4

**Summary:**

This paper introduce a new “parallel recognition → sequential understanding” framework for MLLMs. Specifically, the bottom LLM layers are utilized for parallel recognition and the recognition results are relayed into the top LLM layers for sequential understanding. The parallel recognition in the bottom LLM layers is implemented via object queries, a popular mechanism in DETR. The authors conduct experiments on popular understanding MLLM tasks.

**Strengths:**

1. The proposed method is interesting that transfers  visual recognition and understanding in a same sequential manner to “parallel recognition → sequential understanding” manner.
2.The presentation is good and the figures are well-designed for understanding the proposed methods.

**Weaknesses:**

1. Missing many experiments. This paper propose a framework for both detection and understanding. However, the authors only report the performance of results on VQA MLLMs. They only report inference time and  fps on RefCoCo and Flickr30k, not performance.
2. The compared methods are not newly proposed and current SOTAs, should add more recent methods for comparison, such as LLaVA 1.5, LLaVA-next and etc.

**Questions:**

What is the performance of the proposed method on Flickr30k and RefCOCO?

**Limitations:**

Yes

---

> ### Author Rebuttal · Authors · 2024-08-07
>
> **Q1**: Missing many experiments. This paper propose a framework for both detection and understanding. However, the authors only report the performance of results on VQA MLLMs. They only report inference time and fps on RefCoCo and Flickr30k, not performance.
> **A1**: Table 1 and Section 4.2 in the manuscript already provide performance (acc.) on RefCOCO, as well as other two datasets, i.e., RefCOCO+ and RefCOCOg. These are the three most popular datasets for the referring expression comprehension (REC) task.
> We did not employ Filcker30k because its annotation does not satisfy the standard REC requirement, i.e., binding the bounding box with the corresponding object. No prior methods employed Flicker30k for REC evaluation yet either.
> Given this limitation, during rebuttal, we compute the Rank-1 accuracy as the alternative evaluation protocol, by computing the accuracy between the ground truth boxes and predicted boxes pairs of IoU > 0.5. We observe that our method improves the baseline, e.g.,  +2.4 (59.3 to 61.7) R1 score on Flicker30k test set.
>
> | Method  | val @ R1 | test @ R1 |
> | ------- | -------- | --------- |
> | Shikra  | 58.9     | 59.3      |
> | Octopus | 61.2     | 61.7      |
>
> ---
>
> **Q2**: The compared methods are not newly proposed and current SOTAs, should add more recent methods for comparison, such as LLaVA 1.5, LLaVA-next and etc.
> **A2**: The requested LLaVA-1.5 result is already listed in Table 2 in the manuscript ("LLaVA"). We will revise the notations from "LLaVA" to "LLaVA-1.5" to avoid confusion. Table 2 in the manuscript clearly shows that our method surpasses LLaVA-1.5, e.g., +0.7 on VQAv2 and +1.3 on GQA.
>
> We did not compare LLaVA-Next because it is not an official academic paper, but rather a technical blog. It does not release its training data and code, either.  Since our method is not restricted to any specific MLLM baseline, it can be plugged into LLaVA-Next and is potential to achieve some improvement, as well.
>
> ---
>
> **Q3**: What is the performance of the proposed method on Flickr30k and RefCOCO?
> **A3**: As explained in the responses to Q1, Table 1 in the manuscript already evaluated RefCOCO (+2.01 compared to Shikra baseline).
> Flicker30k is not quite usable for the standard REC task, so we use an alternative evaluation protocol and observe consistent improvement (58.9 $\rightarrow$ 61.2 rank-1 accuracy). Please refer to the response to Q1 for details.

---

> > ### Comment · Reviewer_aS3o · 2024-08-13
> >
> > Thanks for authors' detailed reply. My most concerns have been well addressed, and I would raise my rating as weak accept.

---

> > > ### Author Response · Authors · 2024-08-13
> > >
> > > Thank you for your valuable reviews. We sincerely appreciate you for increasing the rating and recommending acceptance. We are glad that our response addressed your concerns. Your suggestions have greatly contributed to improving our paper, and we will continue to work on and refine our manuscript in the final version.

---

### Official Review · Reviewer_hBAZ · 2024-07-18

**Soundness:** 3
**Presentation:** 3
**Contribution:** 3
**Rating:** 6
**Confidence:** 3

**Summary:**

Octopus first identify unifying visual recoginition and understanding is not optimal from two aspects, 1) parallel recognition could achieve better effiency, 2) recognition results can help understanding. Based on this insight, this work built a recognition then understanding pipeline for MLLM.  Extensive experiments are conducted to show the effectiveness of this work. MLLM achieves better accuracy on common MLLM tasks and is much faster on visual grounding tasks.

**Strengths:**

1. this work give thorough investigation about the connection between visual recognition and understanding in MLLMs.

2.  i like the flow of writing in this work,  the proposed framework is originated from the observed insights, which is quite straightforward yet effective.

3. Extensive experiments show Octopus improves inference efficiency and enhances the accuracy, demonstrating the effectiveness of this work.

**Weaknesses:**

1. More experimental investigations shall be counducted to support the motivation of this paper, especially more quantitative analysis.
2. DETR framework is too old for visual recognition, it is more proper to introduce some efficient architecture for this part, e.g deformable-detr [1]
3. the writing of this work can still be improved, and typos still exists.

[1] Zhu, Xizhou, et al. "Deformable detr: Deformable transformers for end-to-end object detection." arXiv preprint arXiv:2010.04159 (2020).

**Questions:**

see weakness part.
if authors could address my concerns, i would be happy to raise my rating.

**Limitations:**

yes,

---

> ### Author Rebuttal · Authors · 2024-08-07
>
> **Q1**: More experimental investigations shall be conducted to support the motivation of this paper, especially more quantitative analysis.
> **A1**: Thanks. During rebuttal, we conducted one more experiment according to your suggestion. This experiment along with the already-existing ones in Table 1 to 4 in the manuscript can well support our motivation, i.e., the "parallel recognition $\rightarrow$ sequential understanding'' hierarchy brings mutual benefits between recognition and understanding, and is efficient. The details are as below:
>
> **Existing experiments**. Table 1 and Table 4 in the manuscript show that our method improves open-world recognition (REC) accuracy (+2.01 on RefCOCO) and efficiency (5$\times$ faster). Table 2 and Table 3 show the recognition results then benefit the understanding (+0.7 on VQAv2).
>
> **The new experiment in rebuttal**. The improvement on understanding task seems relatively small in Table 2 and Table 3, mainly because many understanding evaluation objective is not closely related to recognition. Hereby, we investigate some closely-related recognition tasks. We selected a subset (173621/214354) of the VQAv2 dataset consisting of "What?" (87868), "How many?" (23372), and "Is?" (62381) type questions, where grounding is needed to recognize the referred objects. We compared the results of our methods against the baseline methods on this subset.
>
> | Method  | LLM Params | Res | VQAv2 |
> | ------- | ---------- | --- | ----- |
> | LLaVA   | 7B         | 336 | 79.3  |
> | Shikra  | 7B         | 224 | 78.8  |
> | Octopus | 7B         | 224 | 81.0  |
>
> As shown in the above table, the advantage of Octopus on this subset is more apparent, which further validate our motivation that parallel recognition can further improves the sequential understanding of MLLM.
>
> ---
>
> **Q2**: DETR framework is too old for visual recognition, it is more proper to introduce some efficient architecture for this part, e.g deformable-detr.
> **A2**: Thanks. During rebuttal, we  substituted the DETR backbone with Deformable-DETR pretrained on COCO-2017 dataset, and trained Octopus with Deformable-DETR on LLaVA-Instruct, REC and Flickr30k datasets. Results are evaluated on RefCOCOg val and test sets, as shown in the table below. We observe further improvement, i.e., +0.5 and +0.4 on validation and test set, respectively. Moreover, during training , we observed that Deformable-DETR significantly accelerates training in earlier stage. We will release  the code for both DETR and the deformable DETR implementations.
>
> | backbone        | val  | test |
> | --------------- | ---- | ---- |
> | DETR            | 79.5 | 80.2 |
> | Deformable-DETR | 80.3 | 81.1 |
>
> ---
>
> **Q3**: The writing of this work can still be improved, and typos still exists.
> **A3**: Thank you for your advice, we will continue to improve the writing and fix typos in the revised version.

---

> > ### Comment · Reviewer_hBAZ · 2024-08-12
> >
> > Thanks for authors' detailed reply. After reading it, the majority of my concerns have been addressed, thus i decide to raise my rating as weak accept.

---

> > > ### Author Response · Authors · 2024-08-13
> > >
> > > We sincerely thank you for the thoughtful feedback and positive rating. We're glad that our responses have addressed your concerns. Your feedback has been invaluable in helping us improve our paper. We appreciate your suggestions and will continue to revise and improve the paper in future versions.

---

### Author Rebuttal · Authors · 2024-08-07

We thank all the reviewers for their valuable comments. We are grateful to Reviewer Up36 and Reviewer aS3o for recognizing the novelty of our method, and to Reviewer hBAZ and Reviewer u3Ag for acknowledging its effectiveness and straightforwardness. In the rebuttal, we provide detailed responses to each reviewer’s feedback. We look forward to your further comments. Thank you.

---

### Decision · Program_Chairs · 2024-09-25

**Decision:**

Accept (poster)

**Comment:**

Unlike previous MLLMs which operate in the sequence-to-sequence fashion, this work proposes a new "parallel recognition -> sequential understanding" framework to work better for visual recognition and understanding tasks. The parallel recognition is implemented via object queries, motivated from DETR, and the outputs are forwarded to the latter sequential understanding.

All reviewers like the idea of "parallel recognition -> sequential understanding" framework and the paper is easy to read. And the main concerns are 1) more experiments need to be conducted to better support the claims and for more fair comparison; 2) the overall improvement over LLava is marginal; 3) limited novelty given some existing works, e.g. "LLava-grounding". After the rebuttal, most of the concerns are resolved, and all reviewers lean to accept. Please follow the reviewers' suggestion to prepare the final version.